# Rutile in Amphibolite Facies Metamorphic Rocks: A Rare Example from the East Qinling Orogen, China

Changming Wang [1,*], Shicheng Rao [1], Kangxing Shi [1], Leon Bagas [1,2], Qi Chen [1], Jiaxuan Zhu [1], Hongyu Duan [1] and Lijun Liu [1]

1. State Key Laboratory of Geological Processes and Mineral Resources, China University of Geosciences, No. 29 Xueyuan Road, Haidian District, Beijing 100083, China; 2001180053@cugb.edu.cn (S.R.); shikx@email.cugb.edu.cn (K.S.); leon.bagas@me.com (L.B.); dirk@cugb.edu.cn (Q.C.); 3001190005@cugb.edu.cn (J.Z.); 3001200042@cugb.edu.cn (H.D.); 2001200052@cugb.edu.cn (L.L.)
2. Xi'an Center of China Geological Survey, 438 Youyi Road, Xi'an 710054, China
* Correspondence: wangcm@cugb.edu.cn; Tel.: +86-10-8232-3761; Fax: +86-10-8232-1006

**Abstract:** Rutile is an important ore mineral to meet the increasing demand of critical metal Ti in various sectors. Here we report a rare example of rutile deposits hosted within the Baishugang–Wujianfang amphibolite-facies metamorphic rocks in the East Qinling Orogen, central China. The rutiles are mostly located within or along the margins of biotite and show 94.6 to 99 wt% $TiO_2$. Rutiles occur as chains, thin layers along the foliation, and dense clusters. The grains are coexisted with magnetite. Based on Zr-in-rutile thermometer the estimated crystallisation temperature is at 630 °C at 7.0 kba. Based on Cr/Nb ratio, the source of the rutile is correlated with Ti-bearing silicate minerals such as biotite from aluminous sedimentary protoliths. The rutile deposit formed during lower amphibolite-facies metamorphism, and is distinct from the eclogite- and granulite-related types elsewhere in the orogen. The LA-ICP-MS U–Pb analyses of rutiles from the deposit yield lower intercept $^{238}U/^{206}Pb$ ages of 386 ± 16 Ma at the Baishugang–Wujianfang district. These ages correspond to a Devonian arc–continent collisional event between the South and North Qinling domains in the East Qinling Orogen.

**Keywords:** rutile deposit; U–Pb geochronology; pressure–temperature conditions; amphibolite-facies metamorphism; East Qinling Orogen

## 1. Introduction

Metamorphic rutile deposits are an important source for Ti, a critical metal supporting the world's increasing demand for various commodities [1,2]. Previous studies have shown that the source rocks for metamorphic and placer rutile deposits are those which have undergone high-pressure and high-temperature metamorphism such as eclogite, retrograde eclogite, and garnet amphibolite [3–5]. Although greenschist- to amphibolite-facies metamorphism plays a major role in Ti enrichment at relatively low pressures and temperatures, the relationship between these lower metamorphic grades and formation of rutile is poorly investigated [6–8]. In this study, we investigate a rare example of rutile deposits hosted within upper greenschist- to lower amphibolite-facies metamorphic rocks in the Baishugang–Wujianfang deposits of East Qinling Orogen (EQO) in central China [9].

Over 50 Mt of potentially extractable titanium metal has been estimated in the Baishugang–Wujianfang districts, making the region the most endowed with rutile in China [10,11]. Previous studies of rutile deposits in this orogen focused on their geology and petrography [12–14], but little is known about the genesis of the deposits. Thus, a more integrated study is necessary for a better understanding of the lithological, geochemical, and geochronological controls.

This paper reports new petrographic data of the mineralisation, LA-ICP-MS rutile and zircon U–Pb ages, and Zr-in-rutile thermometry of the Baishugang–Wujianfang districts.

The genesis of the deposits is discussed with reference to lower amphibolite-facies metamorphism, ore-forming processes, and lithological controls on the mineralisation. We also evaluate our results in the light of mineral exploration aiming to discover large-scale rutile deposits hosted by greenschist- to amphibolite-facies rocks elsewhere in the EQO.

## 2. Geology of the Rutile Deposits in the Baishugang-Wujianfang District

The East Qinling Orogen and adjacent areas include the southern part of the North China Block, the North Qinling Domain and South Qinling Domain (Figure 1). The Shangdan Suture separates the North and South Qinling domains, and the Kuanping Suture separates the North Qinling Domain and the southern part of the North China Block (Figure 1). Both of the North and South Qinling domains host rutile deposits (Figure 1). From north to south, the North Qinling Domain includes the Neoproterozoic Kuanping, Paleozoic Erlangping, Paleoproterozoic Qinling and Meso- to Neoproterozoic Danfeng groups, and Meso- to Neoproterozoic Songshugou Complex. The Kuanping Group consists of greenschist- to amphibolite-facies rocks with tholeiitic basalt or ophiolitic protoliths, mica-schist, gneiss, meta-clastic rocks and marble.

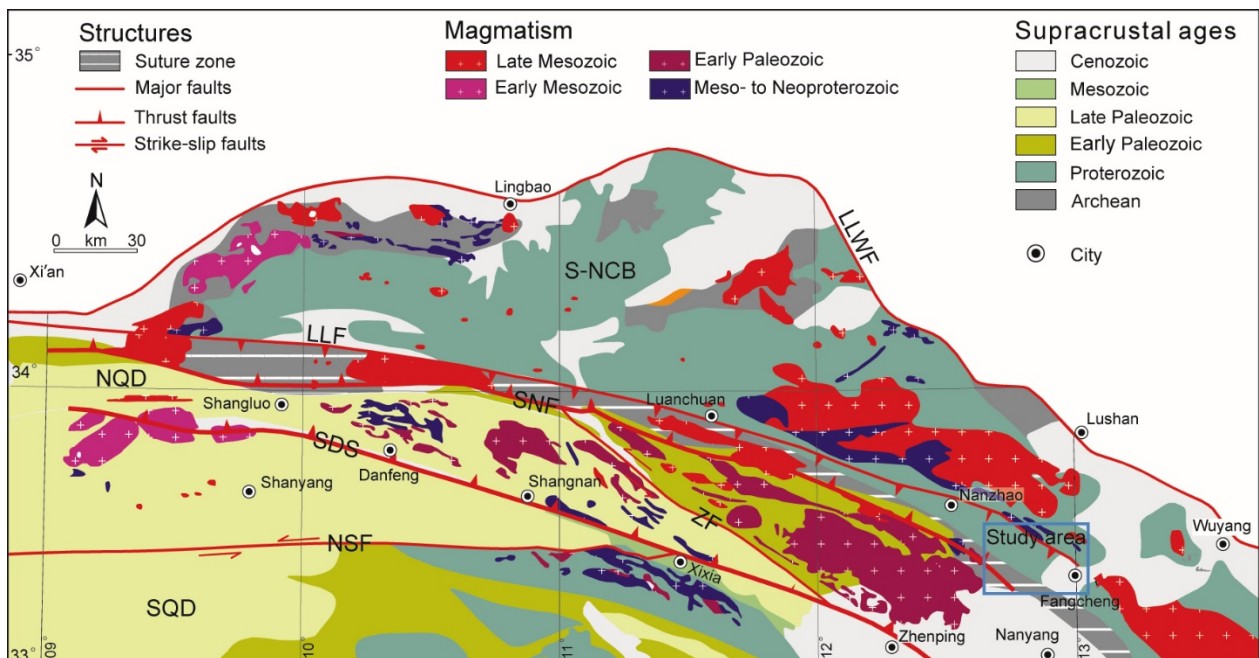

**Figure 1.** Geological map of the Eastern Qinling Orogen showing major rutile deposits. Reprinted with permission from ref. [9]. 2017 Elsevier. Abbreviations: LLWF = Lingbao–Lushan–Wuyang Fault, LLF = Luonan–Luanchuan Fault, SDS = Shangdan Suture, SNF = Shangzhou–Nanzhao Fault, NSF = Ningshan–Shanyang Fault, ZF = Zhuyangguan Fault, NQD = North Qinling Domain, SQD = South Qinling Domain, and S-NCB = Southern North China Block.

The Baishugang–Wujianfang rutile deposits are in the East Qinling Orogen, central China [15]. The Kuanping Group hosts these rutile deposits, which metamorphosed at greenschist- to amphibolite-facies [15]. The estimated total reserve is 600 Mt with a grade of 2.2 wt% $TiO_2$ [16].

The mineralisation in the area is predominantly hosted by gneiss, and to a lesser extent, by biotite schist and gneissic amphibolite (Figure 2). Individual orebodies are several thousand metres long, and several tens to hundreds of metres thick. The mineralisation includes rutile and rutile-bearing silicate minerals as gangue including biotite, hornblende, and diopside. The rutile is translucent and brownish red to dark brown in colour, coexists with magnetite, and forms 0.2 mm long subhedral and euhedral crystals, aggregates, chains or thin layers parallel to the curved foliation (Figures 3A,B and 4A–C).

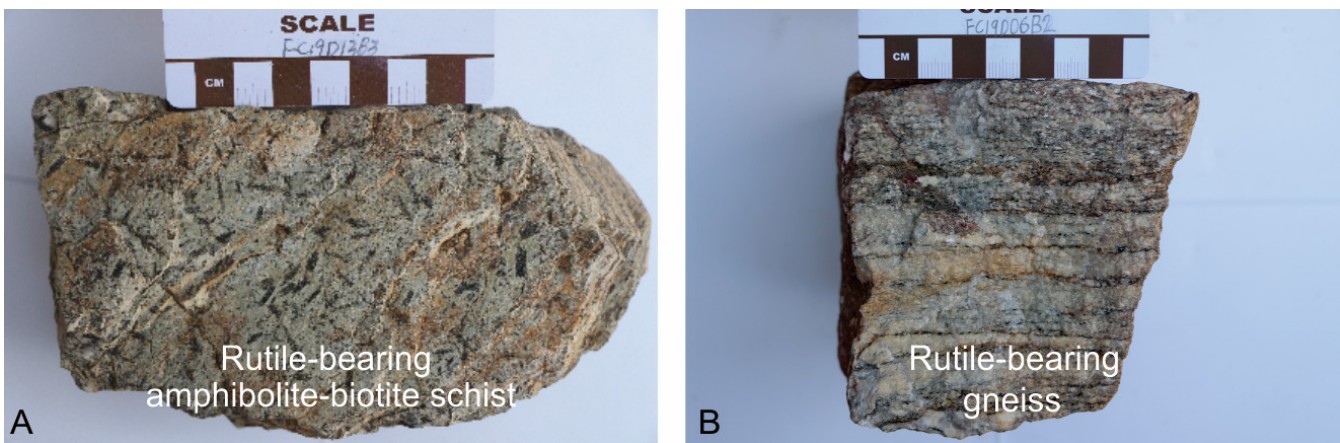

**Figure 2.** Hand specimen photographs at the Bamiao–Qingshan district. (**A**) Rutile-bearing amphibole-biotite schist; (**B**) rutile-bearing gneiss.

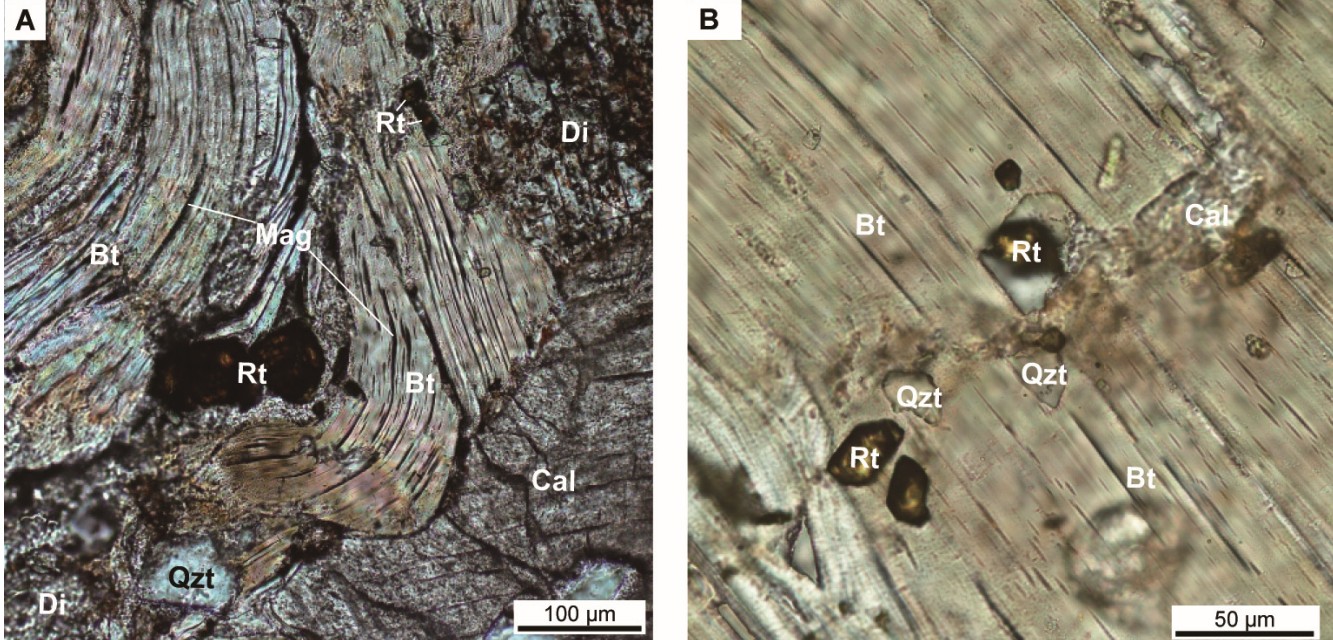

**Figure 3.** Representative photomicrographs under plane-polarized light. (**A**) Dense rutile clusters along the curved foliation formed predominantly by biotite; (**B**) isolated rutile grain in biotite. Abbreviations: Bt = biotite, Cal = calcite, Di = diopside, Mag = magnetite, Qzt = quartz, Rt = rutile.

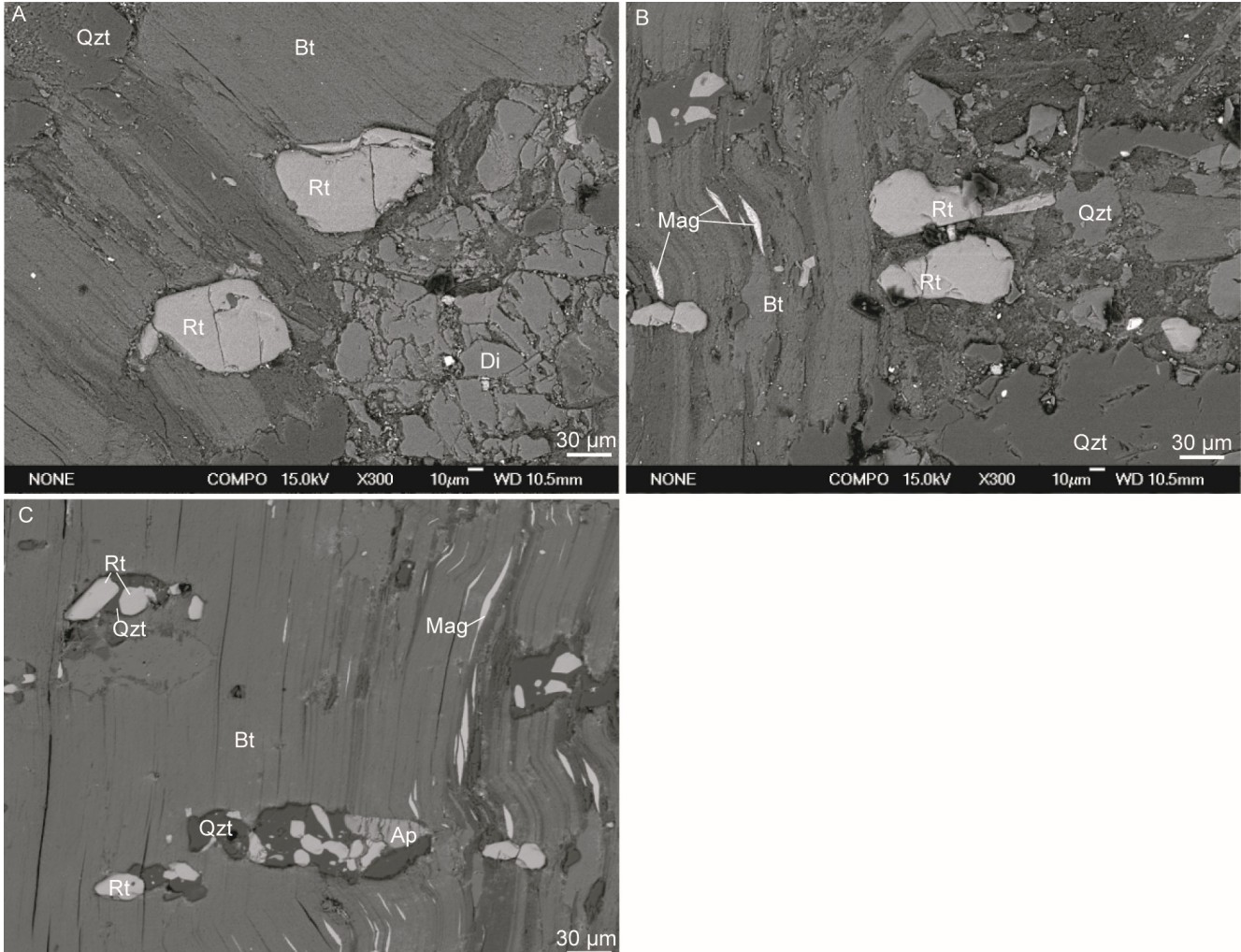

**Figure 4.** Representative backscattered electron SEM images. (**A**) Isolated rutile grain in biotite; (**B**) rutile and magnetite aggregates growing in or around biotite; (**C**) rutile and quartz aggregates, and magnetite growing in biotite. Abbreviations: Ap = apatite, Bt = biotite, Cal = calcite, Di = diopside, Mag = magnetite, Qzt = quartz, Rt = rutile.

## 3. Analytical Methods

### 3.1. Rutile-Bearing Samples

A series of rutile-bearing gneiss, biotite schist and gneissic amphibolite samples in the Kuanping Group were collected from the Baishugang–Wujianfang district for polished thin-sectioning and petrographic studies, electron microprobe analysis (EMPA), and laser ablation–inductively coupled plasma–mass spectrometry (LA-ICP-MS) for zircon and rutile U–Pb dating. In particular, sample FC19D06B2 was selected for EMPA test, and sample FC19D14B2 was selected for LA-ICP-MS zircon and rutile U–Pb dating. For example, the gneiss from the Kuanping Group is dark grey, with the crystallographic axes of the grains oriented in similar directions defining the schistosity, and consists of 30 vol% calcite, 5–20 vol% quartz, 5 vol% rutile and magnetite, and 2 vol% accessory minerals such as zircon and apatite (Figure 3).

### 3.2. Electron Microprobe Analysis

Scanning Electron Microscope (SEM) backscattered electron (BSE) imaging and geochemical analyses of representative mineral grains were performed using a JEOL JXA-8800R Electron Microprobe Analyzer (EMPA). The microprobe was set at a voltage of 20 kV, a beam current of 100 nA, and a 5-μm-wide beam spot at China University of Geoscience,

Beijing (CUGB). The EMPA standards included andradite for Si and Ca, rutile for Ti, corundum for Al, hematite for Fe, eskolaite for Cr, rhodonite for Mn, bunsenite for Ni, periclase for Mg, albite for Na, and K-feldspar for K. All mineral formulas were recalculated using the MINPET 2.0 software [17].

### 3.3. Zircon LA-ICP-MS U–Pb Dating

Samples were selected for LA–ICP–MS zircon U–Pb dating at the Geological Laboratory Center, China University of Geosciences, Beijing. The instrument couples a quadrupole ICP–MS (Agilent 7500a, Santa Clara, CA, USA) and an UP–193 Solid–State laser (193 nm, New Wave). The analyses were performed using a laser spot size measuring 25 μm in diameter, a laser energy density of 8.5 J/cm$^2$, and a repetition rate of 10 Hz. The ablated sample was carried into the ICP–MS in a high-purity helium gas. Calibrations for the zircon analyses were carried out using the NIST 610 glass as an external standard and Si as internal standard. The U–Pb isotope fractionation effects were corrected using the 1064 Ma Zircon 91500 external standard [18]. The 600 Ma GJ-1 zircon standard was also used as a secondary standard to detect deviations on age measurements and calculations [19]. Isotopic ratios and element concentrations of zircons were calculated using the GLITTER 4.4 software from Macquarie University, Australia, following the method by [19]. The common Pb was corrected using the LA–ICP–MS Common Lead Correction v. 3.15 software, following the method outlined by [20]. The uncertainty for individual analyses is quoted at the 1σ level, and the errors on weighted mean ages are at the 95% confidence level.

### 3.4. Rutile LA-ICP-MS U–Pb Dating

Samples of ore were selected for LA–ICP–MS rutile U–Pb dating at the Nanjing Hongchuang Geological Exploration Technology Service Company. The rutiles from the FC19D14B2 sample were mounted in epoxy discs, thin sectioned and polished to expose the grains, cleaned ultrasonically in ultrapure water, then cleaned again prior to the analysis using AR grade methanol. Pre-ablation was conducted for each spot analysis using 5 laser shots (~0.3 μm in depth) to remove potential surface contamination. The analysis was performed using 30 μm diameter spots at 5 Hz with a fluence of 3 J/cm$^2$. Detailed tuning parameters are provided by [21]. The LA-ICP-MS tuning was performed using a 50 μm diameter line scan at 3 μm/s on the NIST 612 glass at ~3.5 J/cm$^2$ with a repetition rate of 10 Hz. The gas flow was adjusted until the sensitivity of the $6 \times 10^5$ cps was reached for the $^{238}$U analyses and the lowest ThO/Th oxide ratio of <0.2% was reached.

The conversion parameters for the analogue and pulse signals in the mass spectrum (P/A) were calibrated on the NIST 610 glass standard using a 100 μm diameter line scan. Other laser parameters are the same as that of the tuning. The mass analyses were completed for $^{49}$Ti, $^{51}$V, $^{53}$Cr, $^{90}$Zr, $^{93}$Nb, $^{120}$Sn, $^{121}$Sb, $^{178}$Hf, $^{181}$Ta, $^{202}$Hg, $^{204}$Pb, $^{206}$Pb, $^{207}$Pb, $^{208}$Pb, $^{232}$Th and $^{238}$U, with a total sweep time of ~0.23 seconds, and the data was reduced using the Lolite software package [22]. The 1758.4 ± 9.7 Ma (2σ) Rutile RMJG was used as primary reference material [23], and double analyses of the RMJG standard were bracketed between multiple groups of 10 to 12 sample unknowns. Typically, 35–40 seconds of the sample signals were acquired after 20 seconds of the gas background analyses. The NIST 610 used as a primary reference material and the BCR basalt standard was used as a secondary reference material. The internal standard was the $^{49}$Ti isotope and the content was assumed to be stoichiometrical. Common Pb uncorrected data were used to construct a Tera-Wasserburg plot to calculate the lower intercept and U–Pb age. The $^{207}$Pb-based common Pb correction method was used for single spot age calculations [24].

## 4. Results

### 4.1. Electron Microprobe Analysis

The composition of the major elements in rutile, biotite and quartz was analysed using the EMPA at CUGB (Table 1).

**Table 1.** Electron microprobe analysis (EMPA) of the major elements (wt%) in rutile, biotite, and quartz.

| Spot No. | Minerals | $SiO_2$ | $TiO_2$ | $Al_2O_3$ | FeO | MnO | MgO | CaO | $Na_2O$ | $K_2O$ | $P_2O_5$ | Total |
|---|---|---|---|---|---|---|---|---|---|---|---|---|
| FC19D06B2 | | | | | | | | | | | | |
| FC19D06B2-1 | Rutile | 0.06 | 98.96 | 0.09 | 0.50 | 0.04 | - | 0.09 | 0.04 | 0.02 | - | 99.80 |
| FC19D06B2-2 | Rutile | 0.03 | 98.09 | 0.08 | 0.38 | - | 0.02 | 0.12 | 0.03 | 0.02 | 0.02 | 98.78 |
| FC19D06B2-3 | Rutile | 0.02 | 97.86 | 0.06 | 0.59 | - | 0.01 | 0.24 | 0.01 | 0.03 | - | 98.83 |
| FC19D06B2-7 | Rutile | 0.03 | 97.89 | 0.03 | 0.44 | 0.05 | - | 0.57 | 0.05 | 0.00 | - | 99.07 |
| FC19D06B2-8 | Rutile | 0.03 | 98.05 | 0.08 | 0.53 | - | 0.01 | 0.24 | 0.04 | 0.09 | 0.03 | 99.09 |
| FC19D06B2-9 | Rutile | 0.23 | 94.59 | 0.83 | 0.63 | - | 0.26 | 1.77 | 0.03 | 0.29 | 0.05 | 98.67 |
| FC19D06B2-11 | Rutile | 0.04 | 97.18 | 0.09 | 0.66 | 0.03 | 0.01 | 0.32 | 0.05 | 0.04 | - | 98.40 |
| FC19D06B2-12 | Rutile | 0.05 | 96.45 | 0.08 | 0.59 | - | 0.01 | 0.53 | 0.05 | 0.12 | 0.01 | 97.90 |
| FC19D06B2-13 | Rutile | 0.05 | 96.94 | 0.05 | 0.64 | - | - | 0.15 | 0.02 | 0.06 | 0.04 | 97.96 |
| FC19D06B2-14 | Rutile | 0.10 | 95.00 | 0.08 | 0.71 | 0.04 | 0.01 | 0.81 | 0.07 | 0.12 | - | 96.93 |
| FC19D06B2-15 | Rutile | 0.02 | 98.07 | 0.07 | 0.33 | - | 0.01 | 0.03 | 0.03 | 0.04 | - | 98.59 |
| FC19D06B2-16 | Rutile | 0.02 | 97.63 | 0.07 | 0.32 | - | 0.00 | 0.07 | 0.04 | 0.05 | - | 98.19 |
| FC19D06B2-17 | Rutile | 0.22 | 97.87 | 0.05 | 0.62 | - | 0.04 | 0.06 | 0.04 | 0.05 | 0.03 | 98.97 |
| FC19D06B2-18 | Rutile | 0.23 | 97.36 | 0.08 | 0.52 | 0.03 | - | 0.04 | 0.04 | 0.05 | - | 98.35 |
| FC19D06B2-19 | Rutile | 0.15 | 97.87 | 0.05 | 0.50 | 0.02 | - | 0.05 | 0.03 | 0.04 | - | 98.70 |
| FC19D06B2-21 | Rutile | 0.17 | 98.17 | 0.06 | 0.36 | - | 0.01 | 0.05 | 0.06 | 0.08 | 0.01 | 98.97 |
| FC19D06B2-22 | Rutile | 0.17 | 98.27 | 0.08 | 0.29 | 0.06 | - | 0.06 | 0.03 | 0.05 | - | 99.00 |
| FC19D06B2-23 | Rutile | 0.10 | 98.74 | 0.06 | 0.36 | - | 0.01 | 0.07 | 0.01 | 0.04 | - | 99.40 |
| FC19D06B2-24 | Rutile | 0.28 | 97.59 | 0.06 | 0.44 | - | 0.01 | 0.24 | 0.04 | 0.04 | - | 98.69 |
| FC19D06B2-28 | Rutile | 0.11 | 98.40 | 0.07 | 0.48 | - | 0.03 | 0.20 | 0.00 | 0.07 | 0.01 | 99.38 |
| FC19D06B2-29 | Rutile | 0.14 | 97.57 | 0.06 | 0.45 | 0.04 | - | 0.16 | 0.02 | 0.07 | - | 98.52 |
| FC19D06B2-31 | Rutile | 0.07 | 98.02 | 0.06 | 0.52 | 0.02 | 0.01 | 0.15 | 0.01 | 0.09 | 0.04 | 98.98 |
| FC19D06B2-32 | Rutile | 0.07 | 97.24 | 0.07 | 0.48 | - | 0.01 | 0.19 | 0.02 | 0.04 | - | 98.10 |
| FC19D06B2-37 | Rutile | 0.28 | 97.76 | 0.10 | 0.31 | 0.03 | 0.02 | 0.16 | 0.04 | 0.04 | 0.05 | 98.79 |
| FC19D06B2-38 | Rutile | 0.05 | 98.16 | 0.09 | 0.29 | 0.06 | - | 0.09 | 0.01 | 0.06 | 0.01 | 98.81 |
| FC19D06B2-39 | Rutile | 0.01 | 97.79 | 0.07 | 0.36 | 0.03 | - | 0.10 | 0.04 | 0.04 | - | 98.43 |
| FC19D06B2-40 | Rutile | 0.03 | 97.45 | 0.08 | 0.26 | 0.04 | - | 0.10 | 0.05 | 0.05 | 0.06 | 98.11 |
| FC19D06B2-4 | Biotite | 38.69 | 1.33 | 17.48 | 7.60 | 0.06 | 19.49 | 0.06 | 0.15 | 9.20 | - | 94.06 |
| FC19D06B2-5 | Biotite | 29.64 | 0.39 | 22.11 | 9.21 | 0.01 | 26.47 | 0.06 | 0.05 | 0.40 | - | 88.33 |
| FC19D06B2-6 | Biotite | 39.14 | 1.22 | 17.71 | 7.48 | 0.03 | 19.18 | 0.09 | 0.11 | 9.30 | - | 94.25 |
| FC19D06B2-10 | Biotite | 38.95 | 1.34 | 18.48 | 7.22 | - | 19.21 | 0.06 | 0.28 | 9.49 | - | 95.01 |
| FC19D06B2-33 | Biotite | 37.03 | 1.22 | 17.72 | 8.38 | - | 19.08 | 0.11 | 0.14 | 9.83 | 0.03 | 93.54 |
| FC19D06B2-34 | Biotite | 37.33 | 1.17 | 17.61 | 8.40 | 0.04 | 19.66 | 0.09 | 0.11 | 9.67 | 0.01 | 94.07 |
| FC19D06B2-35 | Biotite | 37.47 | 1.15 | 17.75 | 8.01 | 0.01 | 19.68 | 0.07 | 0.11 | 9.42 | 0.07 | 93.72 |
| FC19D06B2-36 | Biotite | 37.68 | 1.17 | 17.80 | 8.04 | - | 19.85 | 0.14 | 0.15 | 9.41 | - | 94.22 |
| FC19D06B2-20 | Quartz | 99.36 | 0.24 | 0.03 | 0.26 | 0.05 | - | 0.02 | 0.03 | 0.02 | - | 100.00 |
| FC19D06B2-25 | Quartz | 99.81 | 0.30 | - | 0.19 | - | 0.01 | 0.03 | 0.04 | 0.04 | - | 100.43 |
| FC19D06B2-26 | Quartz | 99.01 | 0.32 | 0.02 | 0.19 | - | - | 0.02 | 0.02 | 0.03 | - | 99.61 |
| FC19D06B2-27 | Quartz | 99.76 | 0.08 | 0.02 | 0.22 | 0.02 | 0.02 | 0.04 | 0.03 | 0.03 | 0.01 | 100.23 |
| FC19D06B2-30 | Quartz | 98.40 | 0.32 | 0.04 | 0.09 | 0.06 | 0.04 | 0.06 | 0.01 | 0.03 | - | 99.04 |

The rutile from the Baishugang–Wujianfang district shows 94.59–98.96 wt% $TiO_2$, 0.03–0.41 wt% $Cr_2O_3$, and 0.26–0.71 wt% FeO. Biotite from the sample has 0.39–1.34 wt% $TiO_2$, 0.01–0.12 wt% $Cr_2O_3$, 7.22–9.21 wt% FeO, and 29.64–39.14 wt% $SiO_2$, and the quartz contains 0.08–0.32 wt% $TiO_2$ and 98.40–99.81 wt% $SiO_2$. The very low concentrations (<0.28%) of $SiO_2$ in the rutiles indicate that the EMPA data collected are not contaminated by zircon inclusions.

### 4.2. Zircon U–Pb Dating

The zircon LA-ICP-MS U–Pb isotope data are reported in Table 2. Twenty-three spots were analysed on zircons from the gneiss, showing 9 to 693 ppm U, 9 to 987 ppm Th, and Th/U ratios of 0.05 to 1.53 (Table 2). The data yield spot ages between 794 and 376 Ma (Figure 5B). The concordant zircons define major age populations at 400–370 Ma, and 790–720 Ma, and the $^{206}Pb/^{238}U$ ages are between 750 and 370 Ma (Figure 5A). The CL images show that most of the zircons have core–rim textures, with cores dated at 790–720 Ma that show oscillatory zoning with high Th/U ratios (>0.4). These Neoproterozoic ages for the cores are in agreement with recently published zircon U–Pb dates by [25]. The rims yield Paleozoic ages of 400–370 Ma and low Th/U ratios of 0.04–0.11 interpreted as metamorphic overgrowths.

**Table 2.** LA–ICP–MS zircon U–Pb analyses.

| Spot No. | Content (ppm) | | | Th/U | Isotopic Ratios | | | | | | | | Ages (Ma) | | | | | |
|---|---|---|---|---|---|---|---|---|---|---|---|---|---|---|---|---|---|---|
| | Pb | Th | U | | $^{207}Pb/^{206}Pb$ | $(\pm1\sigma)$ | $^{207}Pb/^{235}U$ | $(\pm1\sigma)$ | $^{206}Pb/^{238}U$ | $(\pm1\sigma)$ | $^{208}Pb/^{232}U$ | $(\pm1\sigma)$ | $^{207}Pb/^{206}Pb$ | $(\pm1\sigma)$ | $^{207}Pb/^{235}U$ | $(\pm1\sigma)$ | $^{206}Pb/^{238}U$ | $(\pm1\sigma)$ |
| FC19D14B2 | | | | | | | | | | | | | | | | | | |
| 1 | 16 | 9 | 197 | 0.05 | 0.05940 | 0.00700 | 0.50579 | 0.05886 | 0.06176 | 0.00117 | 0.01909 | 0.00284 | 582 | 267 | 416 | 40 | 386 | 7 |
| 5 | 22 | 20 | 289 | 0.07 | 0.05811 | 0.00684 | 0.51294 | 0.05987 | 0.06401 | 0.00096 | 0.01984 | 0.0024 | 534 | 267 | 420 | 40 | 400 | 6 |
| 7 | 25 | 30 | 268 | 0.11 | 0.06747 | 0.00825 | 0.57528 | 0.06961 | 0.06184 | 0.00111 | 0.01884 | 0.00149 | 852 | 267 | 461 | 45 | 387 | 7 |
| 12 | 37 | 76 | 433 | 0.18 | 0.05610 | 0.00222 | 0.51853 | 0.01927 | 0.06703 | 0.00093 | 0.02087 | 0.00034 | 457 | 90 | 424 | 13 | 418 | 6 |
| 16 | 42 | 21 | 547 | 0.04 | 0.04983 | 0.00977 | 0.43163 | 0.08432 | 0.06282 | 0.00098 | 0.01984 | 0.03964 | 187 | 363 | 364 | 60 | 393 | 6 |
| 17 | 73 | 639 | 584 | 1.09 | 0.05749 | 0.00501 | 0.61783 | 0.05331 | 0.07794 | 0.00098 | 0.02419 | 0.00021 | 510 | 198 | 488 | 33 | 484 | 6 |
| 18 | 25 | 87 | 218 | 0.40 | 0.06323 | 0.00117 | 0.87271 | 0.02407 | 0.09906 | 0.00178 | 0.03915 | 0.00092 | 716 | 30 | 637 | 13 | 609 | 10 |
| 20 | 77 | 617 | 409 | 1.51 | 0.06673 | 0.00098 | 1.16028 | 0.01737 | 0.12580 | 0.00106 | 0.03807 | 0.0006 | 829 | 18 | 782 | 8 | 764 | 6 |
| 23 | 22 | 139 | 123 | 1.13 | 0.06682 | 0.00334 | 1.14350 | 0.05584 | 0.12411 | 0.00132 | 0.03786 | 0.00029 | 832 | 107 | 774 | 26 | 754 | 8 |
| 24 | 32 | 162 | 252 | 0.64 | 0.06689 | 0.00148 | 0.93839 | 0.02431 | 0.10136 | 0.00149 | 0.03466 | 0.00066 | 834 | 30 | 672 | 13 | 622 | 9 |
| 25 | 16 | 111 | 95 | 1.17 | 0.07211 | 0.00466 | 1.15088 | 0.06979 | 0.11585 | 0.00134 | 0.03614 | 0.00108 | 989 | 105 | 778 | 33 | 707 | 8 |
| 26 | 94 | 630 | 576 | 1.10 | 0.06634 | 0.00112 | 1.10602 | 0.02417 | 0.12037 | 0.00176 | 0.03857 | 0.00086 | 817 | 23 | 756 | 12 | 733 | 10 |
| 27 | 60 | 386 | 370 | 1.04 | 0.06854 | 0.00509 | 1.05763 | 0.07550 | 0.11192 | 0.00231 | 0.03404 | 0.00056 | 885 | 158 | 733 | 37 | 684 | 13 |
| 28 | 18 | 112 | 169 | 0.66 | 0.05857 | 0.00397 | 0.67301 | 0.04257 | 0.08334 | 0.00204 | 0.02581 | 0.00055 | 551 | 153 | 523 | 26 | 516 | 12 |
| 29 | 20 | 147 | 134 | 1.10 | 0.06156 | 0.00689 | 0.83471 | 0.09205 | 0.09834 | 0.00186 | 0.03028 | 0.00038 | 659 | 251 | 616 | 51 | 605 | 11 |
| 30 | 49 | 381 | 250 | 1.53 | 0.07028 | 0.00184 | 1.28977 | 0.04305 | 0.13105 | 0.00148 | 0.04278 | 0.00096 | 937 | 50 | 841 | 19 | 794 | 8 |
| 31 | 76 | 450 | 431 | 1.04 | 0.06626 | 0.00154 | 1.16074 | 0.02717 | 0.12655 | 0.00120 | 0.04001 | 0.00067 | 815 | 33 | 782 | 13 | 768 | 7 |
| 34 | 55 | 245 | 358 | 0.68 | 0.06505 | 0.00118 | 1.08021 | 0.02023 | 0.12037 | 0.00142 | 0.03940 | 0.00057 | 776 | 21 | 744 | 10 | 733 | 8 |
| 35 | 25 | 121 | 158 | 0.77 | 0.06302 | 0.00272 | 1.04814 | 0.04328 | 0.12062 | 0.00149 | 0.03704 | 0.00038 | 709 | 94 | 728 | 21 | 734 | 9 |
| 36 | 24 | 136 | 170 | 0.80 | 0.06735 | 0.00138 | 1.02357 | 0.02595 | 0.11001 | 0.00196 | 0.03696 | 0.00073 | 849 | 26 | 716 | 13 | 673 | 11 |
| 37 | 31 | 163 | 210 | 0.77 | 0.06397 | 0.00141 | 0.98519 | 0.02233 | 0.11169 | 0.00179 | 0.03945 | 0.00079 | 741 | 23 | 696 | 11 | 683 | 10 |
| 38 | 25 | 127 | 163 | 0.78 | 0.06757 | 0.00136 | 1.09838 | 0.02366 | 0.11825 | 0.00168 | 0.03675 | 0.00074 | 855 | 23 | 753 | 11 | 720 | 10 |
| 39 | 114 | 987 | 693 | 1.42 | 0.06709 | 0.00095 | 1.05011 | 0.01771 | 0.11339 | 0.00163 | 0.03410 | 0.00068 | 840 | 16 | 729 | 9 | 692 | 9 |

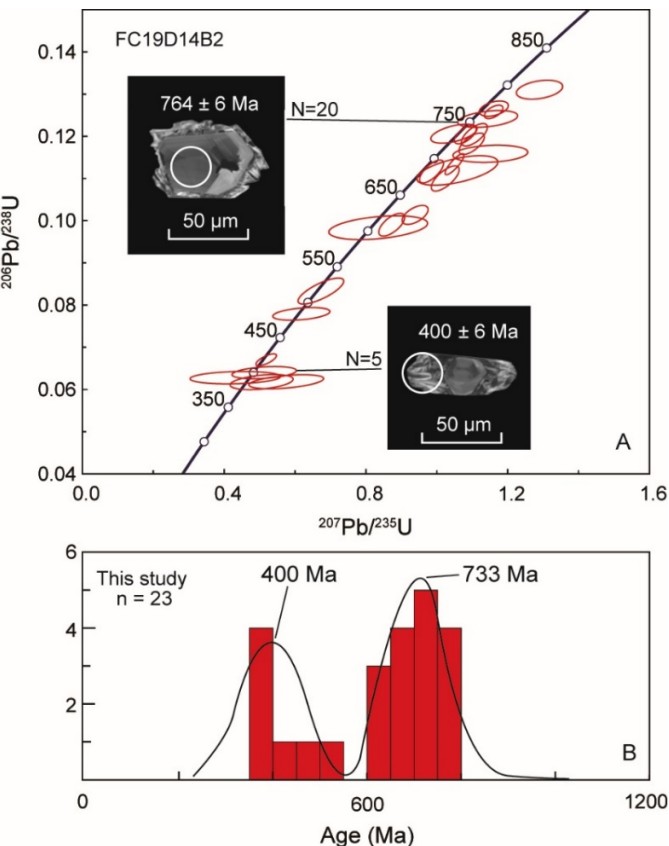

**Figure 5.** Geochronology diagrams: (**A**) Zircon U–Pb concordia plot and representative cathodolumi-nescence (CL) images; and (**B**) frequency (bars) and probability density distribution of LA-ICP-MS dates for detrital zircons from a metasedimentary unit at the Baishugang–Wujianfang district (the error ellipses represent 2σ uncertainties). The location spots for the LA-ICP-MS U–Pb measurements are indicated by the white circles with ages above them.

### 4.3. Rutile U–Pb Dating and Trace Elements

The rutile U–Pb concordia diagrams for the rutile-bearing gneiss from the Baishugang–Wujianfang district are shown in Figure 6.

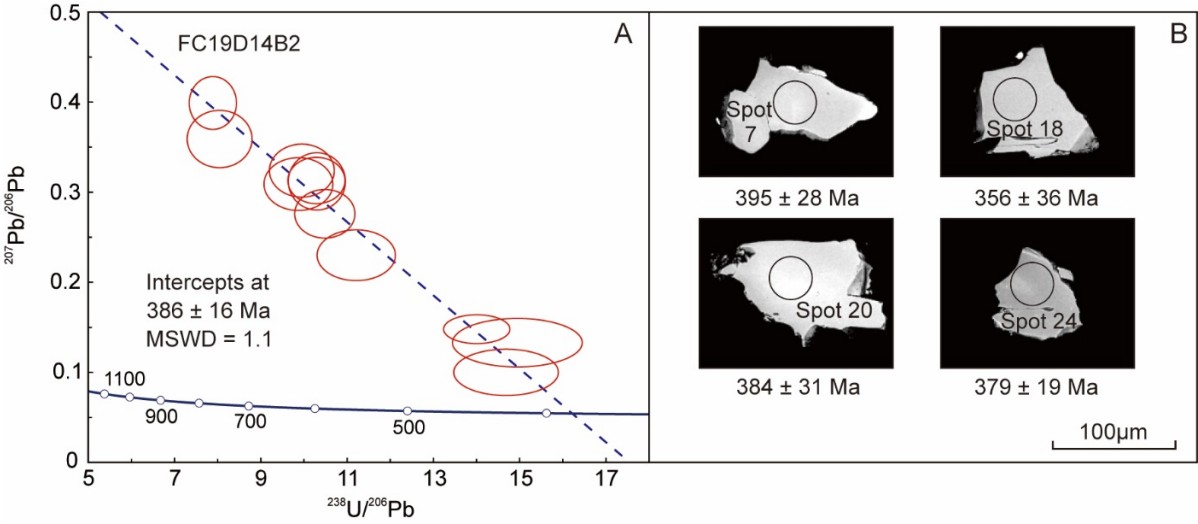

**Figure 6.** Tera–Wasserburg concordia plots for LA-ICP-MS data (**A**) and BSE images (**B**) of rutile in Baishugang–Wujianfang district. The location spots for the LA-ICP-MS U–Pb measurements are indicated by the black circles with ages below them.

The rutile from the gneiss has a relatively uniform U content of 1–7 ppm, but a variable value of $f_{206}$ between 7 and 65% (Table 3). Regression of the data points on the Tera–Wasserburg plot gives a lower intercept age of 386 ± 16 Ma with a MSWD value of 1.1, and the upper intercept with a $^{207}Pb/\,^{206}Pb$ ratio of 0.72 ± 0.52 for the common Pb composition (Figure 6A). Applying the $^{207}Pb$-based common Pb correction method of [24], a weighted mean $^{206}Pb/^{238}U$ age of 374 ± 9 Ma (MSWD = 0.9) is obtained, which is consistent with the lower intercept age within error.

**Table 3.** LA-ICP-MS rutile U–Pb analyses.

| Spot No. | U (ppm) | Th/U | $f_{206}$ [&] (%) | $^{238}U$ [#]$/^{206}Pb$ | ±2σ (%) | $^{207}Pb$ [#]$/^{206}Pb$ | ±2σ (%) | $t_{206/238}$ * (Ma) | ±2σ (Ma) |
|---|---|---|---|---|---|---|---|---|---|
| FC19D14B2 | | | | | | | | | |
| 7 | 2.6 | 0.004 | 7 | 14.6843 | 0.99 | 0.1000 | 0.02 | 395 | 28 |
| 8 | 4.3 | 0.002 | 13 | 14.9477 | 1.23 | 0.1330 | 0.02 | 366 | 33 |
| 9 | 4.0 | 0.175 | 41 | 9.8717 | 0.65 | 0.3090 | 0.02 | 374 | 40 |
| 14 | 3.4 | 0.088 | 56 | 7.8864 | 0.45 | 0.3990 | 0.02 | 353 | 41 |
| 16 | 3.2 | 0.094 | 49 | 8.0451 | 0.61 | 0.3590 | 0.03 | 396 | 53 |
| 18 | 3.8 | 0.026 | 42 | 10.2987 | 0.54 | 0.3120 | 0.03 | 356 | 30 |
| 20 | 3.4 | 0.059 | 36 | 10.4822 | 0.57 | 0.2760 | 0.02 | 384 | 31 |
| 21 | 4.1 | 0.024 | 43 | 9.9502 | 0.61 | 0.3240 | 0.02 | 356 | 36 |
| 22 | 4.4 | 0.045 | 42 | 10.2881 | 0.54 | 0.3130 | 0.02 | 355 | 30 |
| 23 | 3.7 | 0.027 | 28 | 11.2108 | 0.74 | 0.2300 | 0.02 | 400 | 36 |
| 24 | 5.6 | 0.002 | 15 | 14.0056 | 0.63 | 0.1480 | 0.01 | 379 | 19 |

Note: [&] $f_{206}$ is the percentage of common $^{206}Pb$ in total $^{206}Pb$, calculated by $^{207}Pb$-based. * $t_{206/238}$ is $^{206}Pb$–$^{238}U$ age calculated by $^{207}Pb$-based common-lead correction. [#] The ratios are common Pb uncorrected, used for Tera–Wasserburg plot.

The LA–ICP–MS analyses of trace elements and estimated crystallisation temperatures of the rutile are listed in Table 4. The rutile from the Baishugang–Wujianfang district shows 89–773 ppm Zr, 15,000–64,500 ppm Nb, 1434–4090 ppm Ta, and 89–443 ppm Cr.

**Table 4.** LA–ICP–MS trace element analyses (ppm) of rutile deposits in the study area and their estimated temperature (°C) of crystallisation.

| Spot No. | V | Cr | Zr | Nb | Sn | Sb | Hf | Ta | W | T [1] | T [2] | T [3] |
|---|---|---|---|---|---|---|---|---|---|---|---|---|
| FC19D14B2 | | | | | | | | | | | | |
| 7 | 343 | 22 | 89 | 50,190 | 193 | 458 | 5 | 2042 | 837 | 553 | 564 | 589 |
| 8 | 336 | 66 | 118 | 61,300 | 226 | 441 | 6 | 4090 | 771 | 572 | 600 | 609 |
| 14 | 298 | 15 | 311 | 49,840 | 207 | 457 | 10 | 2135 | 653 | 645 | 724 | 685 |
| 16 | 270 | 9 | 124 | 43,210 | 235 | 513 | 5 | 1255 | 674 | 576 | 606 | 613 |
| 18 | 267 | 30 | 142 | 42,270 | 255 | 520 | 7 | 1434 | 810 | 586 | 624 | 623 |
| 21 | 362 | 60 | 262 | 64,500 | 214 | 409 | 9 | 3618 | 798 | 632 | 702 | 671 |
| 22 | 219 | 7 | 209 | 30,690 | 189 | 340 | 7 | 1333 | 524 | 614 | 673 | 652 |
| 23 | 199 | 12 | 145 | 19,660 | 176 | 357 | 6 | 1757 | 260 | 587 | 626 | 624 |
| 24 | 188 | 10 | 113 | 15,000 | 182 | 266 | 5 | 2365 | 161 | 569 | 594 | 606 |
| Average ± St. dev. | | | | | | | | | | 593 ± 29 | 634 ± 50 | 630 ± 30 |

Note: [1,2,3] Temperatures were estimated according to the thermometry by [26–28] at 7–8 kbar, respectively.

## 5. Discussion

### 5.1. Protolith for Rutile Deposits and Titanium Sources

Rutile is a known host for high-field-strength elements (HFSEs), such as Nb, Ta and Cr, which are widely used to fingerprint the geochemical processes in the Earth's mantle and crust [5,29,30].

The Cr and Nb contents of rutile allow discrimination between various sources for rutile, such as metapelite and metamafic rocks [29]. Rutile, however, is extremely compatible with Nb compared to most metamorphic phases, and concentrates >90% of the available Nb and Ti in a metamorphic rock; consequently, the Nb/Ti ratio of the source rock is mirrored in the rutile [31].

Studies [5,29–31] pointed out that rutile samples with Cr < Nb and accompanied by Nb > 800 ppm was derived from a metapelite source, whereas rutile with Cr < Nb and Nb < 800 ppm was derived from a metamafic source. The log (Cr/Nb) value of 0 is

the boundary between metapelite and metamafic sources for rutile, where positive log (Cr/Nb) values are indicative of a metamafic source, although this method is not reliable for rutile with low Cr and Nb concentrations [32]. The rutile individuals in our samples hold >800 ppm Nb, with the lowest value being 15,000 ppm Nb. The Cr content is less than that of Nb (Figure 7). This indicates that the rutile from the Baishugang–Wujianfang districts derived from metapelitic sources, which is consistent with previous studies [5,29–31].

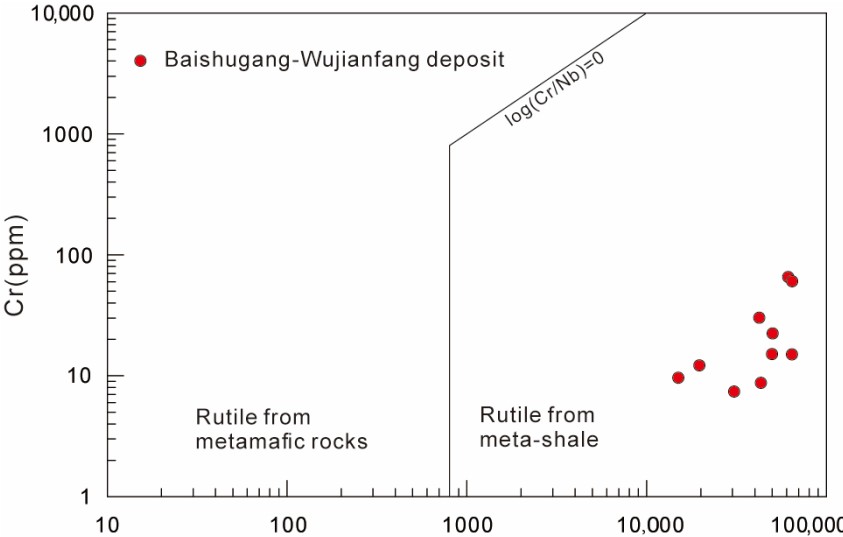

**Figure 7.** Cr versus Nb discrimination diagram for rutiles from the Baishugang–Wujianfang districts. Reprinted with permission from ref. [33]. 2018 Springer.

The reflected light petrological studies and BSE images show that the rutile crystals are mainly concentrated within biotite (Figures 3 and 4). The rutile shows 94.59–98.96 wt% $TiO_2$, the biotite shows 0.39–1.34 wt% $TiO_2$ and quartz assays 0.08–0.32 wt% $TiO_2$ (Table 1). Previous studies reported that biotite and pyroxene are rich in titanium, and the possible mechanism for rutile enrichment could be decomposition of Ti-rich biotite [2,5]. The biotite contains high $TiO_2$ content (Table 1), and the rutile occurs along the curved foliation formed predominantly by biotite in this study (Figure 3A). Thus, the titanium sources of rutile are possibly related to rutile-bearing silicate minerals such as biotite from metapelitic rocks in the study area.

*5.2. Crystallisation Temperatures and Possible Metamorphic Path*

Away from tectonic plate boundaries, the temperature increases by about 25 °C per km depth [31]. The temperature range for the amphibolite-facies is commonly assumed to be 500–750 °C, corresponding to a crustal depth of about 25 km.

The Zr-in-rutile thermometer is commonly regarded as a reliable tool for estimating the temperatures at which zircons crystallise, based on the temperature-dependent replacement of Ti by Zr in the rutile lattice [26–28]. In our studied samples, zircon grains occur within or adjacent to the rutile crystals and quartz forms aggregates in the ore (Figure 4B,C), making possible the application of the Zr-in-rutile geothermometry.

Three types of Zr-in-rutile thermometers have been proposed for the estimation of the rutile crystallisation temperatures with or without pressure calibrations (Table 4):

T (°C) = 127.8 × ln (Zr in ppm) − 10 [26];
T (°C) = (4470 ± 120)/((7.36 ± 0.10) − log (Zr rutile)) − 273 [27];
T (°C) = (85.7 + 0.473 P)/(0.1453 − R ln (Zr in ppm)) − 273 [28].

The estimated crystallisation temperature for rutile from the Baishugang–Wujianfang district is calibrated as the average values of 593 ± 29 °C (n = 9) and 634 ± 50 °C (n = 9) without pressure, and at a pressure of 7.0 kbar, as 630 ± 30 °C (n = 9) [28,34,35] (Table 4).

These temperature estimates are consistent with the metamorphic grade of the region studied (Figure 8). The lower estimated temperatures (630 °C) are consistent with an intermediate P/T ratio [36]. The higher temperatures are consistent with those estimated by conventional thermometers for the lower unit of the Kuanping Group (i.e., at a pressure of 9.3–11.2 kbar and a temperature of 610–570 °C) and the group's upper unit being at a pressure of 6.6–8.9 kbar and a temperature of 650–630 °C [35,37–39]. Figure 8 shows the rutile deposits at the Baishugang–Wujianfang districts are distinct from the metamorphism such as eclogite- and granulite-facies elsewhere in the orogen.

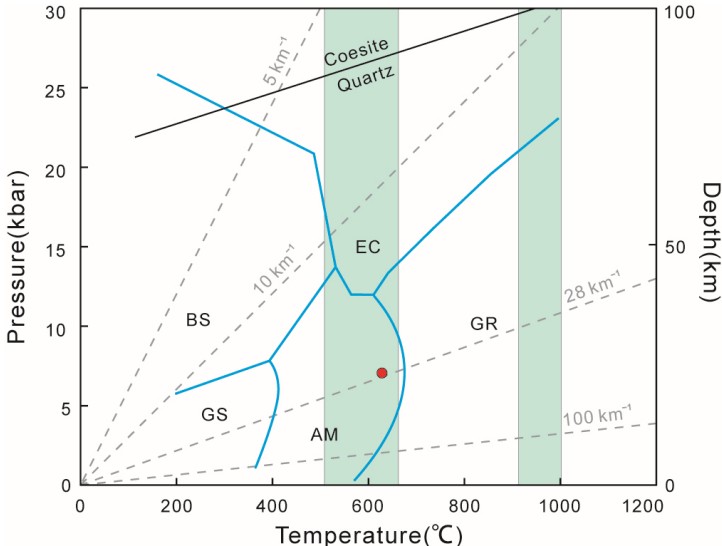

**Figure 8.** Pressure–temperature diagram. Reprinted with permission from ref. [40]. 2002 Wiley Online Library. The red circle represents an average value (630 ± 30 °C) of estimated crystallisation temperature for rutile from the Baishugang–Wujianfang district at a pressure of 7.0 kbar. Abbreviations of metamorphic facies: BS = blueschist-facies, GS = greenschist-facies, AM = amphibolite-facies, GR = granulite-facies, EC = eclogite-facies.

The rutiles from the Baishugang–Wujianfang district coexist with magnetite and have high-Nb contents of 15,000–64,500 ppm (Figures 3 and 4). In addition, no exsolution lamellae of ilmenite and sphene, or chlorite replacing biotite and pyroxene were observed in the samples under reflected light microscopy and in backscattered electron (BSE) images. This indicates that retrograde metamorphism from granulite-facies is not an option.

We thus suggest that most of the rutile at the Baishugang–Wujianfang districts are related to prograde metamorphism at temperatures of around 600 °C corresponding to a crustal depth of ~25 km. The rutile deposits at the Baishugang–Wujianfang districts are distinct from the high-pressure and high-temperature metamorphism such as eclogite- and granulite-related types elsewhere in the orogen [37]. The enrichment of a large amount of rutile formed during prograde metamorphism under medium-temperature and medium-pressure conditions is undoubtedly of great significance for the research of rutile deposit.

### 5.3. Interpretation of U–Pb Dating and Tectonic Setting

The Kuanping Group in the Baishugang–Wujianfang district was metamorphosed to the lower amphibolite-facies [41]. The peak metamorphism in the group has been recently dated at ca. 440 Ma [37–39,42]. In addition, muscovite and biotite from the group yields a well-defined $^{40}Ar/^{39}Ar$ plateau age of 365–383 Ma, interpreted as constraining the age for cooling [43].

The rims of zircon from a metasedimentary rock from the Kuanping Group have Paleozoic ages of 418–386 Ma and low Th/U ratios of 0.04–0.11, interpreted as the metamorphic age of the host rocks. Similar $^{40}Ar/^{39}Ar$ age of about 416 Ma was reported for amphibole at the Baishugang–Wujianfang district [12]. As discussed above, the rims of the rutile from

the same sample yields a La-ICP-MS U–Pb metamorphic age of about 386 Ma (Figure 6A). Together with previous studies [9,44], we therefore suggest that metamorphism in the area took place during the Devonian at about 418–386 Ma.

On the basis of the comparison with zircon, LA-ICP-MS U–Pb analyses of rutile yield homogeneous age, constraining metamorphic event and supplying the upper limit of the regional metamorphism.

## 6. Conclusions

Rutile deposits are hosted in amphibolite-facies metamorphic rocks in the East Qinling Orogen. The source of the rutile is correlated with Ti-bearing silicate minerals from aluminous sedimentary protoliths. The rutile at the Baishugang–Wujianfang district is related to prograde metamorphism at the temperature of about 630 °C with the pressure of 7.0 kba, recording amphibolite-facies metamorphism. The rutile deposit formed during metamorphism of amphibolite-facies, and is distinct from high-pressure and high-temperature metamorphisms such as eclogite- and granulite-related types elsewhere in the orogen. The enrichment of rutile in amphibolite-facies metamorphism is not only of great significance to the prograde metamorphic and metallogenic mechanism of rutile, it also provides an important key with respect to regional prospects of mineral exploration.

LA-ICP-MS U–Pb analyses of rutile yield a lower intercept U–Pb age of $386 \pm 16$ Ma, constraining and supplying the upper limit of the regional metamorphism. Furthermore, rutile U–Pb dating has some advantages compared with zircon, such as the homogeneous age constraining metamorphic event in the Baishugang–Wujianfang district.

**Author Contributions:** Conceptualisation, C.W.; methodology, S.R. and Q.C.; software, S.R.; validation, C.W.; formal analysis, S.R. and Q.C.; investigation, S.R., K.S., J.Z., H.D. and L.L.; resources, C.W.; data curation, C.W. and S.R.; writing—original draft preparation, C.W. and S.R.; writing—review and editing, L.B. and K.S.; visualisation, S.R. and K.S.; supervision, C.W.; project administration, C.W.; funding acquisition, C.W. All authors have read and agreed to the published version of the manuscript.

**Funding:** This research is partially funded by the National Key Research and Development Project of China (Number 2020YFA0714802), the National Natural Science Foundation of China (Number 41872080), and Most Special Fund from the State Key Laboratory of Geological Processes and Mineral Resources in China University of Geosciences, Beijing (CUGB), China (Number MSFGPMR201804).

**Institutional Review Board Statement:** Not applicable.

**Informed Consent Statement:** Not applicable.

**Data Availability Statement:** The datasets presented in this study can be obtained upon request to the corresponding author.

**Acknowledgments:** The authors thank the team members at CUGB for their field support, data analysis, constructive discussions, and comments.

**Conflicts of Interest:** The authors declare no conflict of interest.

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
