# Peer review of "Rutile in Amphibolite Facies Metamorphic Rocks: A Rare Example from the East Qinling Orogen, China"

_applsci, doi:10.3390/app11188756_

Round 1

Reviewer 1 Report

Dear Authors,

I was asked to review the work entitled “Rutile in amphibolite facies metamorphic rocks: A rare example from the East Qinling Orogen, China”, by Changming Wang, Shicheng Rao, Kangxing Shi, Leon Bagas, Qi Chen, Jiaxuan Zhu, Hongyu Duan and Lijun Liu.

The paper focuses on the metamorphic conditions and age for the formation of rutile in deposits and rocks from the Baishugang−Wujianfang district, China. Authors also highlight the importance of broaden the knowledge on the conditions of formation of rutile, with a view to the search and exploitation of Ti deposits.

I found this is an interesting article, quite well-organized, and can be certainly considered for publication in Applied Sciences. Not having direct expertise in Mineralogy, nor with regard to Metamorphic Geology, I am not fully able to judge the correct application of analytical protocols. However, the analyzes described and the results proposed seem to be consistent with literature, and the interpretation justified by the results. In my opinion, the manuscript needs some revision, mainly minor adjustments, which the authors should consider before publication. Line-by-line comments are in the attached file; here, some general comment and suggestion:

1) I am not a native English, so I am not the most suitable in proposing any linguistic and stylistic adjustment. I found the text plain and well-written, although some sentences are not clear, for me at least. For example, see my comments at lines 54-58, 267.

2) Please check the Journal formatting style, especially for citations, both through the main text and in References.

3) In my opinion, it is hard to follow geological setting as well as geological implications without a geological map. Please note not everybody is familiar with the geology of the Region, so some descriptions and logical connections are hard to picture. Moreover, I noted a bit of confusion between geographic and geological terms (e.g., line 267). Geological map and schemes, while simplified or general they could be, would help the reader.  

4) Analytical methods and Results sections are well-structured and explicative. My only concerns are about the number of analyzed samples, and the criteria of choice. The results of analyzes carried out on only two samples are reported: why only two? Are they exemplificative of a larger dataset? On what criteria were they chosen? Which lithostratigraphic units do they come from? At what point in local succession? Is this a preliminary report, to test the methods before proceeding with a wider study? All relevant information, but none is available to the reader. I Think that details should be added, either in the geological settings or in the methodological section. Also note each wide-ranging speculation based on not enough data, or on a limited dataset, can be criticized. I am not strictly an expert on Mineralogy, but I think this is a general rule in Science. There is the need to justify the choice.

5) Most of the criticism is about Discussion and Conclusions sections. I agree with deductions, as all the data point in the direction proposed by the authors. Nonetheless, I think this final part is a weak point for the paper, and it should be reassessed and enlarged. First, the cause-effect relationships, or what is the starting point and what are the inferences, are not always clear to the reader. For example, all data point to metamorphism in amphibolite facies, but is not clear if this is already known (and data confirm the starting hypothesis), or the results allow identifying a metamorphic facies not recognized before. The same ambiguity can be remarked about the individuation of protolith, the T range for crystallization of rutile, and the geological implication for the age of metamorphism: what is the starting point, the data presented here? Or they rather confirm previous literature? Second, all the inferences about the origin of Ti in rutile deriving from biotite should be better constrained. Finally, in the Conclusions section, clearly remark what do authors think are the results of this work, what the implications for the different sectors (enlarged knowledge about rutile crystallization conditions in a metamorphic context? A wider presence of rutile than what supposed in the study area, with geoeconomic implications? New insights for reconstructing the metamorphic history and the geological evolution of the Region? All of them?), what are the research perspectives. My suggestion is to improve sections 5 (particularly 5.3) and 6, clarifying these points, in order to enhance the value of research and strengthen the results.

6) All figures are of good quality and functional to the paper, and only minor adjustments are needed (see my comments). Nonetheless, I do not fully understand the meaning of Figure 7. In this diagram, only one point is reported: what does it mean? If the aim is to associate data to the amphibolite facies field, to insert one only point is not enough. A figure showing the more general distribution among metamorphic facies of rock samples from the study area (including the ones here provided) should be more explicative.   

I hope my comments could be somehow helpful.

Best Regards

Author Response

Explanations for the comments of Reviewer 1 are addressed below:

(1) I am not a native English, so I am not the most suitable in proposing any linguistic and stylistic adjustment. I found the text plain and well-written, although some sentences are not clear, for me at least. For example, see my comments at lines 54-58, 267.

Response to comments:

  • Thanks for your helpful suggestions and comments! We have revised the whole text to correct the linguistic and stylistic problems (see the revised manuscript).

(2) Please check the Journal formatting style, especially for citations, both through the main text and in References.

Response to comments:

  • We have checked and corrected the problems about the formats and citations in the whole text and references (see the revised manuscript).

(3) In my opinion, it is hard to follow geological setting as well as geological implications without a geological map. Please note not everybody is familiar with the geology of the Region, so some descriptions and logical connections are hard to picture. Moreover, I noted a bit of confusion between geographic and geological terms (e.g., line 267). Geological map and schemes, while simplified or general they could be, would help the reader.

Response to comments:

  • Thanks for the good suggestion! Actually, we have already published several articles about this region such as the reference [9] (Wang, C.M.; Deng, J.; Bagas, L.; Wang, Q. Zircon Hf–isotopic mapping for understanding crustal architecture and metallogenesis in the Eastern Qinling Orogen. Gondwana Res. 2017, 50, 293−310). The readers could easily see the simplified geological map of this region from these studies. And this study focuses on the analysis and genesis of rutile, so there is no detailed geological map in the manuscript (see the revised manuscript).

(4) Analytical methods and Results sections are well-structured and explicative. My only concerns are about the number of analyzed samples, and the criteria of choice. The results of analyzes carried out on only two samples are reported: why only two? Are they exemplificative of a larger dataset? On what criteria were they chosen? Which lithostratigraphic units do they come from? At what point in local succession? Is this a preliminary report, to test the methods before proceeding with a wider study? All relevant information, but none is available to the reader. I think that details should be added, either in the geological settings or in the methodological section. Also note each wide-ranging speculation based on not enough data, or on a limited dataset, can be criticized. I am not strictly an expert on Mineralogy, but I think this is a general rule in Science. There is the need to justify the choice.

Response to comments:

  • Actually, a series of rutile-bearing samples were collected from the Baishugang−Wujianfang district for polished thin-sectioning and petrographic studies, electron microprobe analysis (EMPA), and laser ablation-inductively coupled plasma-mass spectrometry (LA-ICP-MS) zircon and rutile U–Pb dating. Especially, sample FC19D14B2 was selected for zircon and rutile U–Pb dating.

(5) Most of the criticism is about Discussion and Conclusions sections. I agree with deductions, as all the data point in the direction proposed by the authors. Nonetheless, I think this final part is a weak point for the paper, and it should be reassessed and enlarged. First, the cause-effect relationships, or what is the starting point and what are the inferences, are not always clear to the reader. For example, all data point to metamorphism in amphibolite facies, but is not clear if this is already known (and data confirm the starting hypothesis), or the results allow identifying a metamorphic facie not recognized before. The same ambiguity can be remarked about the individuation of protolith, the T range for crystallization of rutile, and the geological implication for the age of metamorphism: what is the starting point, the data presented here? Or they rather confirm previous literature? Second, all the inferences about the origin of Ti in rutile deriving from biotite should be better constrained. Finally, in the Conclusions section, clearly remark what do authors think are the results of this work, what the implications for the different sectors (enlarged knowledge about rutile crystallization conditions in a metamorphic context? A wider presence of rutile than what supposed in the study area, with geoeconomic implications? New insights for reconstructing the metamorphic history and the geological evolution of the Region? All of them?), what are the research perspectives. My suggestion is to improve sections 5 (particularly 5.3) and 6, clarifying these points, in order to enhance the value of research and strengthen the results.

Response to comments:

  • Thanks for your careful suggestions! We have revised the related contents. Main contribution in this manuscript: (1) According to rutile, crystallization temperatures of the rutile in amphibolite facies of metamorphic rocks are estimated at 630°C; (2) By the comparison with zircon, LA-ICP-MS U–Pb analyses of rutile yield homogeneous age, constraining metamorphic event and supplying the upper limit of the regional metamorphism. (3) The rutile deposit formed during metamorphism of amphibolite facies, and is distinct from the high-pressure and high-temperature metamorphism such as eclogite and granulite-related types elsewhere in the orogen (see the revised manuscript).

(6) All figures are of good quality and functional to the paper, and only minor adjustments are needed (see my comments). Nonetheless, I do not fully understand the meaning of Figure 7. In this diagram, only one point is reported: what does it mean? If the aim is to associate data to the amphibolite facies field, to insert one only point is not enough. A figure showing the more general distribution among metamorphic facies of rock samples from the study area (including the ones here provided) should be more explicative.

Response to comments:

  • We have revised the figures which are not in good quality and function. The red circle in Figure 7 represents an average value (630° ± 30°C) of estimated crystallisation temperature for rutile from the Baishugang−Wujianfang district at a pressure of 7.0 kbar (see the revised manuscript).
  • Besides, more revision details could be seen in the manuscript. We have adopted most of the comments and suggestions and revised our paper in details according to your helpful revision (see the revised manuscript).

Thank you for the helpful comments and reviews.

All authors

Reviewer 2 Report

Dear Authors, I have read this text with attention. 
I like the clear layout of the paper. The methods and deductions based on available data and other research work were described very well.
However, I would like to ask the authors to rethink their statements from the line 216-2018 versus 225-228. In my opinion, there is a thought dodge here, some simplification that may suggest wrong contradictory results to the reader. I suspect that the authors had in mind the crystallization of rutile at the expense of biotite in the metamorphic process and the ingress of Ti from biotite, which in turn entered biotite from the sedimentary protolith. However, the text in these lines may suggest com different. Please clarify this paragraph.
The second issue is the ending. Conclusions from the whole article are too poor, I suggest some generalization. The authors themselves write that it is a rare case of rutile in amphibole rocks, that the process of rutile crystallization was not precisely known, so please answer these questions in the conclusion. I think that the conclusions should be more elaborated. 
Also, there are minor typos e.g. line 68 and two dots, figure 9 before 7 (you might want to swap the no.), the caption of fig 9 is above the diagram not below. In the case of figure 4, I suggest stretching the diagram a bit - if there is no age above 1Ga, why should the scale be as large as 4Ga?
Apart from that, the text seems to be well written.

Author Response

Explanations for the comments of Reviewer 2 were addressed below:

(1) However, I would like to ask the authors to rethink their statements from the line 216-218 versus 225-228. In my opinion, there is a thought dodge here, some simplification that may suggest wrong contradictory results to the reader. I suspect that the authors had in mind the crystallization of rutile at the expense of biotite in the metamorphic process and the ingress of Ti from biotite, which in turn entered biotite from the sedimentary protolith. However, the text in these lines may suggest come different. Please clarify this paragraph.

Response to comments:

  • Thanks for your careful suggestions! Studies [5, 26−28] pointed out that rutile samples with Cr<Nb and accompanied by Nb>800 ppm derived from a metapelite source, whereas rutile with Cr<Nb and Nb<800 ppm derived from a metamafic source. The rutile individuals in our samples hold >800 ppm Nb, with the lowest value being 15000 ppm Nb. The Cr content is less than that of Nb (Figure 6). This indicates that the rutile in the Baishugang−Wujianfang districts derived from metapelitic sources, which is consistent with previous studies [5,26−28] (see the revised manuscript).

(2) The second issue is the ending. Conclusions from the whole article are too poor, I suggest some generalization. The authors themselves write that it is a rare case of rutile in amphibole rocks, that the process of rutile crystallization was not precisely known, so please answer these questions in the conclusion. I think that the conclusions should be more elaborated.

Response to comments:

  • Thanks for your careful suggestions! The rutile deposit formed during metamorphism of amphibolite facies, and is distinct from the high-pressure and high-temperature metamorphism such as eclogite and granulite -related types elsewhere in the orogen (see the revised manuscript).

(3) Also, there are minor types e.g. line 68 and two dots, figure 9 before 7 (you might want to swap the no.), the caption of figure 9 is above the diagram not below. In the case of figure 4, I suggest stretching the diagram a bit - if there is no age above 1Ga, why should the scale be as large as 4Ga?

Response to comments:

  • We have revised the text and figures (see the revised manuscript and figures).

Thank you for the helpful comments and reviews.

All authors
